# Improved Electrochemical Performance of Sm_0.2_Ce_0.8_O_1.9_ (SDC) Nanoparticles Decorated SrCo_0.8_Fe_0.1_Ga_0.1_O_3−δ_ (SCFG) Fiber, Fabricated by Electrospinning, for IT-SOFCs Cathode Application

**DOI:** 10.3390/ma16010399

**Published:** 2023-01-01

**Authors:** Marzieh Kiani, Mohammad Hossein Paydar

**Affiliations:** Department of Materials Science and Engineering, School of Engineering, Shiraz University, Shiraz 71348-51154, Iran

**Keywords:** electrospinning, infiltration, solid oxide fuel cells, SCFG, cathode

## Abstract

This paper examines the electrochemical and microstructural features of SrCo_0.8_Fe_0.1_Ga_0.1_O_3−δ_ (SCFG) with a fibrous structure infiltrated by an SDC electrolyte for use as a cathode in solid oxide fuel cells (SOFCs). An electrospinning process is used to produce SCFG fibers. In a symmetrical cell, Sm_0.2_Ce_0.8_O_1.9_ (SDC) nanoparticles are infiltrated into the porous fibrous SCFG cathode layer after it was applied to the SDC dense electrolyte. Electrochemical impedance spectroscopy (EIS) analysis reveals that the polarization resistance of the SCFG cathode with fiber morphology is significantly lower than that of the same combination with powder morphology. In addition, it is shown that infiltration of SDC oxygen ion conductor nanoparticles enhanced electrochemical performance. The lowest value of polarization resistance, 0.03 Ω cm^2^ at 800 °C, is attained by the SCFG with a fibrous structure containing 14 wt% SDC nanoparticles.

## 1. Introduction

Energy has long been one of the most crucial needs in human societies. On the other hand, the widespread use of fossil fuels has created several issues, including a rise in global temperature, the destruction of the ozone layer, the production of toxic gases, acid rain, and water and soil pollution. Therefore, the development of clean energy and renewable energy sources is vital for human civilization. Consequently, fuel cells are one of the sources of clean energy production, as they can directly convert chemicals into electrical energy. Among fuel cells, solid oxide fuel cells (SOFCs) exhibit the highest efficiency and excellent flexibility in terms of fuel type [1,2,3,4,5].

Accordingly, it is essential to develop intermediate-temperature solid oxide fuel cells (IT-SOFCs) with high efficiency to reduce system costs and enable commercialization. On the other hand, decreasing the operating temperature increases the polarization resistance of the cathode due to a decrease in oxygen reduction reaction (ORR) activity, thereby reducing the overall efficiency of IT-SOFCs. Therefore, to improve the performance of IT-SOFCs, it is essential to develop cathode materials with excellent catalytic activity for an ORR at low operating temperatures [6,7,8,9,10,11,12].

In addition, the SOFC cathode must possess a high electrical conductivity, thermal expansion compliance, and chemical compatibility with other cell components. Oxides with a defective perovskite structure (ABO_3−d_), including the SrCoO_3−δ_ family, have been introduced as excellent cathode materials for SOFC application due to their high oxygen ion conductivity and superior electrical conductivity [12,13,14,15,16,17,18].

Among these compounds, SrCo_0_._8_Fe_0_._1_Ga_0.1_O_3−δ_ (SCFG) has been introduced as a mixed ionic and electronic conductor (MIEC) with sufficient high electrical conductivity of 122 S·m^−1^ at 800 °C, suitable oxygen permeability, and stability in reducing atmosphere at high temperatures [19,20,21]. The thermal expansion coefficient (TEC) of the SCFG cathode is reported to be 12.6 × 10^−6^ K^−1^ in the temperature range of room temperature to 800 °C, which is comparable to the TEC value obtained for the Sm_0.2_Ce_0.8_O_1.9_ (SDC) electrolyte [19]. As a result of its proper thermal expansion compatibility with electrolytes and superior electrochemical performance, SCFG oxide can be considered a suitable cathode material for IT-SOFCs.

Electrochemical reactions occur at the three-phase boundary (TPB) of the SOFCs’ cathode, whereas for mixed ionic-electronic conductors, it is widely spread throughout the cathode layer. Furthermore, the presence of electrolyte particles in composite cathodes generates a more expanded three-phase boundary, which increases oxygen permeability and enhances the cathode’s catalytic activity [1,22]. In addition to the chemical composition, the performance of a cathode is strongly dependent on the specific surface area of the pores resulting from the microstructure. Therefore, the method of cathode compound preparation and its processing is important and effective on the cathode performance.

Recently, electrospinning has been introduced as a promising technique for producing polymer and ceramic fibers with high specific surface area per unit mass and small-sized porosity [22,23,24,25]. Several cathode compounds, including La_0.6_Sr_0.4_Co_0.2_Fe_0.8_O_3−δ_ [26], Bi_1−x_Sr_x_FeO_3−δ_ (BSFO) [27], Sm_0.5_Sr_0.5_CoO_3−δ_-Gd_0.2_Ce_0.8_O_1.9_ [28], BaCe_0.5_Fe_0.5−x_Bi_x_O_3−δ_ [29], and La_1.6_Sr_0.4_NiO_4_ (LSN) [30], have been fabricated in the form of fibers for use in SOFCs via the electrospinning method. According to different carried out studies, fibrous cathodes exhibit improved electrochemical performance compared to powder morphology. Furthermore, electrode surface modification via solution infiltration has been introduced as an effective technique for achieving the desired microstructure and high catalytic activity. Several studies revealed that the cathode’s oxygen surface exchangeability could be enhanced by adding oxygen ion conductive phase nanoparticles [31,32,33,34].

To this end, several recent studies have shown that composite SOFC cathodes with nanofibrous structures produced by electrospinning plus solution infiltration demonstrate higher electrochemical performance than composite electrodes with powder morphologies. Among them, it can refer to GDC infiltrated LSCF nanofibers [35], La_0.8_Sr_0.2_Co_0.2_Fe_0.8_O_3−δ_ tubes impregnated with Ce_0.8_Gd_0.2_O_1.9_ nanoparticles [36], GDC infiltrated Pr_0.4_Sr_0.6_Co_0.2_Fe_0.7_Nb_0.1_O_3−δ_ (PSCFN) nanofibers [37], and LSM infiltrated YSZ nanofibers [38]. Nano-fibrous structural properties, such as high specific surface area, superior gas permeability, continuous electrical and ionic conduction paths, and high concentration of hetero-interfaces, are responsible for improved electrochemical catalytic activity.

In the present work, the electrospinning method is explored for the fabrication of perovskite oxide SrCo_0.8_Fe_0.1_Ga_0.1_O_3−δ_ (SCFG) fibers. Infiltration of the SCFG fibrous cathodes with different amounts of nano SDC particles is also adopted to fabricate composite cathodes with improved electrochemical properties.

## 2. Experimental Process

### 2.1. Synthesis of SDC Electrolyte Phase and SCFG Fibers and Preparation of Symmetric Cells

SCFG fibers were produced through the electrospinning process. Initially, the required amounts of Sr(NO_3_)_2_, Co(NO_3_)_2_·6H_2_O, Fe(NO_3_)_3_·9H_2_O, and Ga(NO_3_)_3_·xH_2_O (x = 8) (Sigma Aldrich, St. Louis, MO, USA) were dissolved in deionized water to prepare the electrospinning solution based on the stoichiometry of SrCo_0.8_Fe_0.1_Ga_0.1_O_3−δ_ compound. The polyvinyl alcohol (PVA) polymer was subsequently dissolved in deionized water before being added to the clear and homogenous nitrate solution. Afterward, the created homogenous solutions were stirred for 5 h at room temperature using a magnetic stirrer. In this instance, the electrospinning solution concentration was 7 g L^−1^, and the mass ratio of total nitrate powders to PVA was 1:1. The electrospinning solution was then placed in a 10 mL syringe attached to a nozzle with a 21-needle gauge. The electrospinning procedure was then implemented at 27 kV.

The feeding rate of the syringe pump during the electrospinning procedure was 0.7 mL h^−1^, and the distance between the syringe nozzle and the collector plate was 14 cm. The electrospinning process yielded fibers that were piled on aluminum foil placed on the collector. The collected fibers were then calcined for 5 h at 950 °C. The SDC electrolyte powder was produced via the solid-state reaction method by firing a mixture of CeO_2_ and Sm_2_O_3_ powders at 1200 °C for 5 h. The prepared SDC powder was passed through a 60-mesh screen, pressed into green tablets with a diameter of 1 cm and a thickness of approximately 1 mm using 100 MPa pressure in a solid die, and then fired at 1400 °C for 5 h.

In this study, the polarization resistance of cathodes was measured using the electrochemical impedance spectroscopy (EIS) method. For the EIS test, the structure of symmetric cells supported by electrolyte was SCFG-x wt% SDC/SDC/SCFG-x wt% SDC (x = 0, 7, 14, 21). To prepare symmetrical cells, a slurry containing 65 wt% alpha-terpineol solution, 35 wt% SCFG fibers, and ethyl cellulose powder equal to 5 wt% of the amount of spent cathode fibers was used.

A homogeneous cathode slurry was prepared by slowly stirring the mixture with a magnetic stirrer for one hour. The resultant cathode slurry was applied to the sintered SDC electrolyte tablets using a spatula. Multiple applications of the slurry to the electrolyte were required to achieve the desired thickness. The cathodes applied to SDC electrolyte tablets were sintered at a temperature of 1000 °C for 2 h at a heating rate of 3 °C min^−1^. The thickness of the applied electrodes was approximately 25 microns, and the cathode’s effective surface area in the obtained symmetrical cells was 0.79 cm^2^.

SDC nanoparticles were infiltrated into the scaffold of SCFG cathodes to modify their surface. According to the SDC stoichiometry, Sm(NO_3_)_3_·6H_2_O (Sigma Aldrich, USA) and Ce(NO_3_)_3_·6H_2_O (Sigma Aldrich, USA) powders were dissolved in deionized water to prepare the infiltration solution. Citric acid was then added to the solution as a complexing agent, in a molar ratio of 3:1 citric acid: total metal cations. The prepared solution included 0.2 mol L^−1^ of SDC. Afterward, ethanol equal to 30% *v*/*v* of the prepared solution was added to improve the wetting properties of the infiltration solution.

To carry out the infiltration process, the stoichiometric droplets of the prepared solution were placed on the porous layer of the fibrous SCFG cathode using a microliter syringe and they were infiltrated into the porous cathode scaffold by capillary force. SDC nanoparticles were synthesized and coated on the surface of the SCFG cathode after a 1 h firing process at a temperature of 1000 °C. The amount of infiltrated SDC nanoparticles was determined by measuring the difference in weight of the total symmetric cell before and after each solution infiltration and heat treatment. By repeating the solution infiltration/heat treatment process, the concentration of SDC nanoparticles was regulated to approximately 7, 14, and 21 wt%.

### 2.2. Characterization

The PVA/SCFG precursor fibers were characterized using a thermogravimetric-differential thermal analyzer (TG-DTA) with a heating rate of 10 °C min^−1^ within the temperature range of 25–800 °C. The X-ray diffraction (XRD) technique was employed to ensure the formation of SDC and SCFG phases and to check the chemical compatibility between them after heating at 1000 °C for 1 h. The microstructure of the electrospun fibers before/after calcination, as well as different prepared cathodes, were examined using field emission scanning electron microscopy (FESEM). The EIS test of symmetrical cells was conducted at temperatures between 600 and 800 °C in air. The impedance curves under an open circuit were recorded in the frequency range of 0.1–1 MHz, where the applied signal amplitude was 40 mV. Finally, Z-view software (version 3.4) was used to analyze R_p_ data.

## 3. Results and Discussion

### 3.1. Thermal Analysis

The TG-DTA curves of PVA/SCFG precursor fibers from room temperature to 1000 °C are presented in Figure 1. The TG thermogram (dashed line curve) revealed the sample’s weight loss between 50 and 80 °C, which corresponds to the endothermic reaction of physically absorbed water removal. Furthermore, a decrease in sample mass is observed between 200 and 500 °C, correlating with the decomposition and release of PVA polymer. The metal nitrates then undergo phase transformation, resulting in the formation of a perovskite structure. Peaks on the DTA curve also represent these transformations. Simultaneously, the TG curve exhibits a downward trend at the relevant temperatures, indicating a substantial weight loss during the formation of the perovskite phase [39,40,41].

### 3.2. Phase Characterizations

The XRD pattern of the SDC electrolyte powder obtained via the solid-state reaction method (Figure 2a) indicates that it has been formed with a fluorite crystal structure and sufficient purity [42]. The XRD graph of SCFG fibers calcined at 950 °C for 5 h is depicted in Figure 2b. The SCFG composition exhibits the characteristic peaks associated with a perovskite structure [21]. In addition, the XRD pattern for the mixture of SCFG fibers and SDC infiltration solution after 1 h of firing at 1000 °C is depicted in Figure 2c. As can be seen, only the peaks associated with a perovskite structure of the SCFG and the characteristic peaks of the SDC can be observed; no other impurity peaks are present. These results indicate that the SDC precursor was converted into fluorite oxide by calcination at 1000 °C for 1 h without undergoing any chemical reaction with the SCFG parent phase.

### 3.3. Microstructural Studies

Figure 3 illustrates the microscopic image of SCFG fibers produced by electrospinning before/after 5 h calcination at 950 °C. As can be seen, the average diameter of fibers before calcination is 115 nm. In addition, the structure remains continuous and fibrous after calcination and polymer removal (Figure 3b). The scanning electron microscopy (SEM) image of the symmetrical SCFG/SDC/SCFG cell cross-section is shown in Figure 4a. The cathode’s porous microstructure on the dense electrolyte layer and the adhesion between the cathode and electrolyte layers are observable. As is evident, the cathode layer thickness is uniform, ranging between 20 and 30 microns. Figure 4b–e depict the typical section microstructures of SCFG fiber cathodes before/after decoration with SDC nanoparticles at approximately 7, 14, and 21 wt%, respectively.

Before infiltration (Figure 4b), the surface of the SCFG fibrous cathode is porous and smooth. After infiltration of approximately 7 wt% of SDC nanoparticles into the fiber-structured cathode body and subsequent heating at 1000 °C for 1 h, as depicted in Figure 4c, the SDC nanoparticles are observed to be randomly distributed on the surface of the SCFG fibrous cathode with a size of approximately 30 nm. According to the XRD pattern, the SDC precursor transformed successfully into fluorite oxide after calcination at 1000 °C for 1 h. After infiltration of at about 14 wt% SDC nanoparticles into the fibrous structure of the SCFG cathode, microscopic images revealed that the surface of the SCFG cathode had been modified continuously and uniformly. However, loading the cathode with approximately 21 wt% of SDC nanoparticles (Figure 4e) causes the excessively infiltrated particles to aggregate. Moreover, when SDC nanoparticles are overloaded, they are sintered together at high temperatures and create larger grains.

### 3.4. Electrochemical Performance

The polarization resistances of pure and infiltrated fibrous cathodes were evaluated using an electrochemical impedance (AC) spectroscopy test of symmetrical cells under open circuit conditions at temperatures between 600 and 800 °C in air. Figure 5 depicts the Nyquist impedance spectrum of SCFG cathodes in the pure and infiltrated states containing 7, 14, and 21 wt% SDC nanoparticles. In order to clearly show the difference in the cathode’s polarization behavior, ohmic resistances (R-Ohmic) were subtracted from the impedance data and are presented in Table 1. Multiple processes, including charge transfer, oxygen surface and volume penetration, and oxygen absorption-desorption, influence the impedance of a cathode. The charge transfer process is associated with the high-frequency impedance response (R1/CPE-1), whereas the oxygen surface and volume penetration processes, as well as oxygen absorption-desorption, are responsible for the low-frequency impedance response (R2/CPE-2) [42,43,44,45]. As it is difficult to distinguish the impedance contribution of different components, Table 1 just reports the total value of polarization resistance (R_p_) for variously prepared cathodes in the 600–800 °C temperature range.

Table 1 presents the total values of polarization resistance (Rp) and the fitted equivalent circuit parameters for various prepared cathodes in the 650–800 °C temperature range. As the SDC content initially increases, both R1 and R2 decrease, which is an indication of the effect of this second phase on the promotion of the ORR process. It is worth noting that the R2 values played a dominant role in the total Rp, implying that the process of oxygen absorption-desorption was predominantly rate-limiting. However, the R1 value increases when 21 wt% SDC nanoparticles are incorporated into the fibrous SCFG scaffold. Accordingly, the excessive addition of the SDC nanoparticles may restrict the charge transfer process. Moreover, as is evident, the polarization resistance of all prepared cathodes decreases as the temperature increases. Indeed, as the temperature rises, the concentration of oxygen vacancy and electrical conductivity is increased, which leads to a decrease in R_p_ [46].

As expected, the cathode’s electrochemical performance highly depends on its composition and microstructure. Compared to the conventional SCFG cathode results obtained by other researchers [21], changing the morphology of the SCFG cathode from powder to fiber has significantly decreased its polarization resistance. At 750 °C, the polarization resistance of an SCFG powder cathode was determined to be 0.3 Ω cm^2^ [21], whereas this value was obtained as 0.2 Ω cm^2^ for the fibrous SCFG cathode. Comparing the fibrous SCFG cathode to the conventional SCFG cathode, the advantageous microstructure with continuous pores and more reaction sites decreases the polarization resistance of the cathode. At 800 °C, the polarization resistance of pure fibrous SCFG cathode was obtained as 0.12 Ω cm^2^.

According to the data presented in Table 1, decoration of the fibrous SCFG cathode with SDC nanoparticles resulted in a decrease in polarization resistance at various temperatures, indicating a significant improvement in electrochemical activity. The SDC phase is an excellent conductor of oxygen ions, and when the fibrous SCFG cathode is uniformly decorated by SDC nanoparticles, the processes of surface absorption, decomposition, and oxygen transport can be enhanced. A pure SCFG compound is a mixed ionic and electronic conductor, but its oxygen ion conductivity is not high enough at temperatures between 600 and 800 °C. Consequently, in the pure phase of SCFG, the oxygen reduction reaction occurs mainly at and near the triple phase boundary (TPB) of the SCFG cathode/SDC electrolyte/O_2_, whereas the TPB is expanded by adding an SDC electrolyte to all areas of the SCFG cathode. At 800 °C, the R_p_ value decreased to 0.08 Ω cm^2^ when 7 wt% SDC nanoparticles were infiltrated into the porous body of the fibrous SCFG cathode. By increasing the loading of SDC nanoparticles in the fibrous structure of the SCFG cathode to 14 wt%, the lowest polarization resistance values were obtained in the temperature range of 600–800 °C. Due to the continuous porous channels that facilitate gas flow, the polarization resistance of SDC infiltrated fibrous SCFG cathodes is considerably reduced. However, more loading of 21 wt% SDC nanoparticles into the SCFG cathode caused a slight increase in polarization resistance, which can be interpreted as a decrease in cathode porosity and TPB, or nanoparticles agglomeration.

Figure 6 depicts the Arrhenius curve of the polarization resistance for variously prepared cathodes. The activation energy (Ea) for the cathodic reaction was determined to be 1.08 eV for a fibrous SCFG cathode in the temperature range of 600–800 °C, which is relatively lower than the value reported by other researchers [19,21] for a conventional SCFG cathode. The infiltrated cathodes with SDC nanoparticles exhibit slightly lower activation energy than the pure SCFG cathode with a fibrous structure.

## 4. Conclusions

In this research, a novel composite cathode composed of SCFG fibers, fabricated by an electrospinning process, and decorated with SDC nanoparticles through an infiltration process, is fabricated. The results of the phase characterization and thermal analyses revealed the well crystalline phase formation and excellent high-temperature chemical compatibility of the prepared composite cathodes. Microstructural analyses, carried out by high resolution SEM, revealed the creation of well-distributed SDC nanoparticles on the porous body of the fibrous SCFG structure.

The R_p_ value for the fibrous SCFG cathode was determined to be 0.12 Ω cm^2^ at a temperature of 800 °C, which is significantly lower than what was obtained for the same chemical composition cathode, prepared by using SCFG fine powder. On the other hand, it was demonstrated that the infiltration of SDC nanoparticles into the SCFG cathode with a fiber structure increases the active sites and enhances the oxygen reduction reaction (ORR) catalytic activity in the cathode. By infiltrating approximately 7, 14, and 21 wt% SDC nanoparticles into the body of the fibrous SCFG cathode, at 800 °C, the R_p_ value was determined to be 0.08, 0.03, and 0.04 Ω cm^2^, respectively, which is significantly lower than what was obtained for the pure fibrous SCFG cathode. Consequently, it can be concluded that the electrochemical performance of electrospun electrodes may be improved through a simple infiltration technique and heat treatment, which can facilitate the low-cost commercialization of SOFC technology.

## Figures and Tables

**Figure 1 materials-16-00399-f001:**
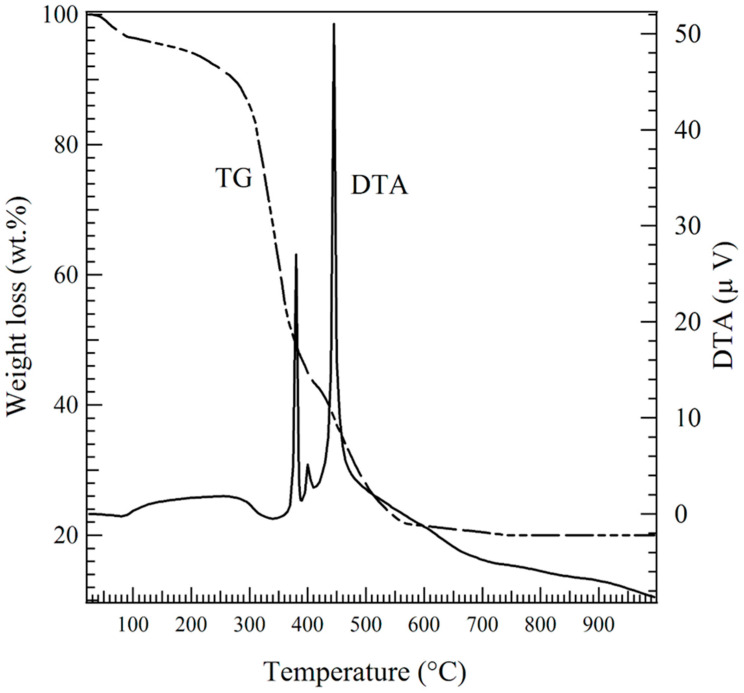
TG-DTA curves of the PVA/SCFG precursor fibers at a heating rate of 10 °C min^−1^.

**Figure 2 materials-16-00399-f002:**
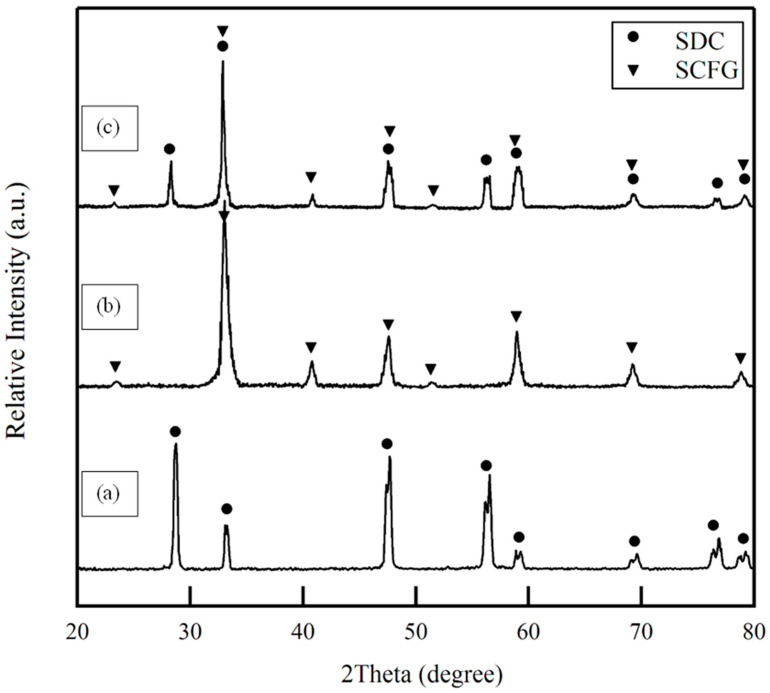
X-ray diffraction pattern of (**a**) SDC electrolyte powder calcined at 1200 °C for 5 h; (**b**) SCFG fibers calcined at 950 °C for 5 h; (**c**) SDC-infiltrated SCFG calcined at 1000 °C for 1 h.

**Figure 3 materials-16-00399-f003:**
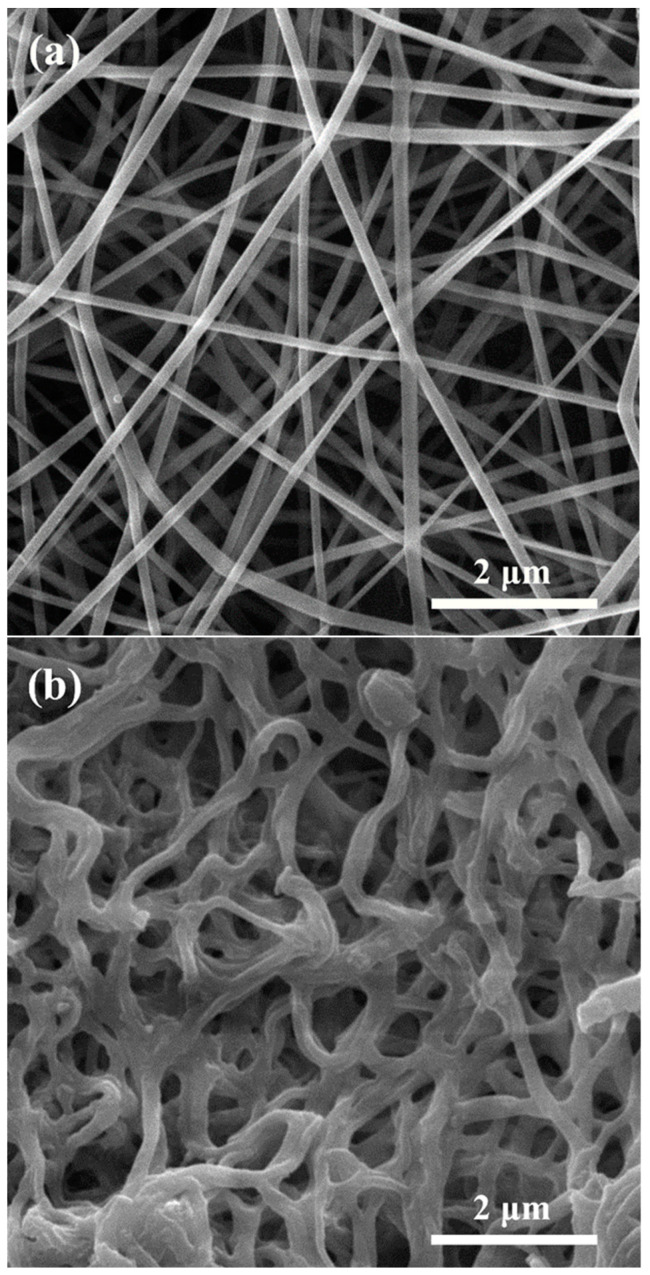
SEM images of (**a**) PVA/SCFG precursor fibers; (**b**) SCFG fibers after calcination at 950 °C for 5 h.

**Figure 4 materials-16-00399-f004:**
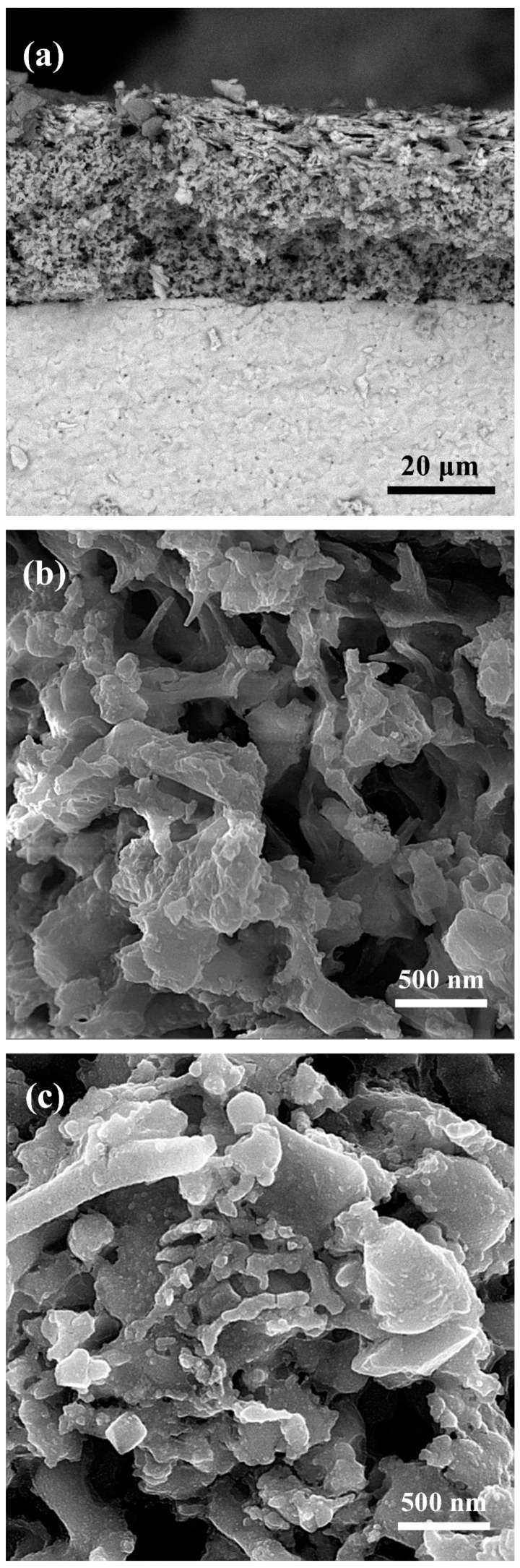
SEM image of (**a**) the cross-section of the SCFG/SDC/SCFG symmetrical cell created by SCFG fibers; (**b**–**e**) the typical section of SCFG cathode with fibrous structure before/after infiltration of about 7, 14, and 21 wt% of SDC nanoparticles, respectively.

**Figure 5 materials-16-00399-f005:**
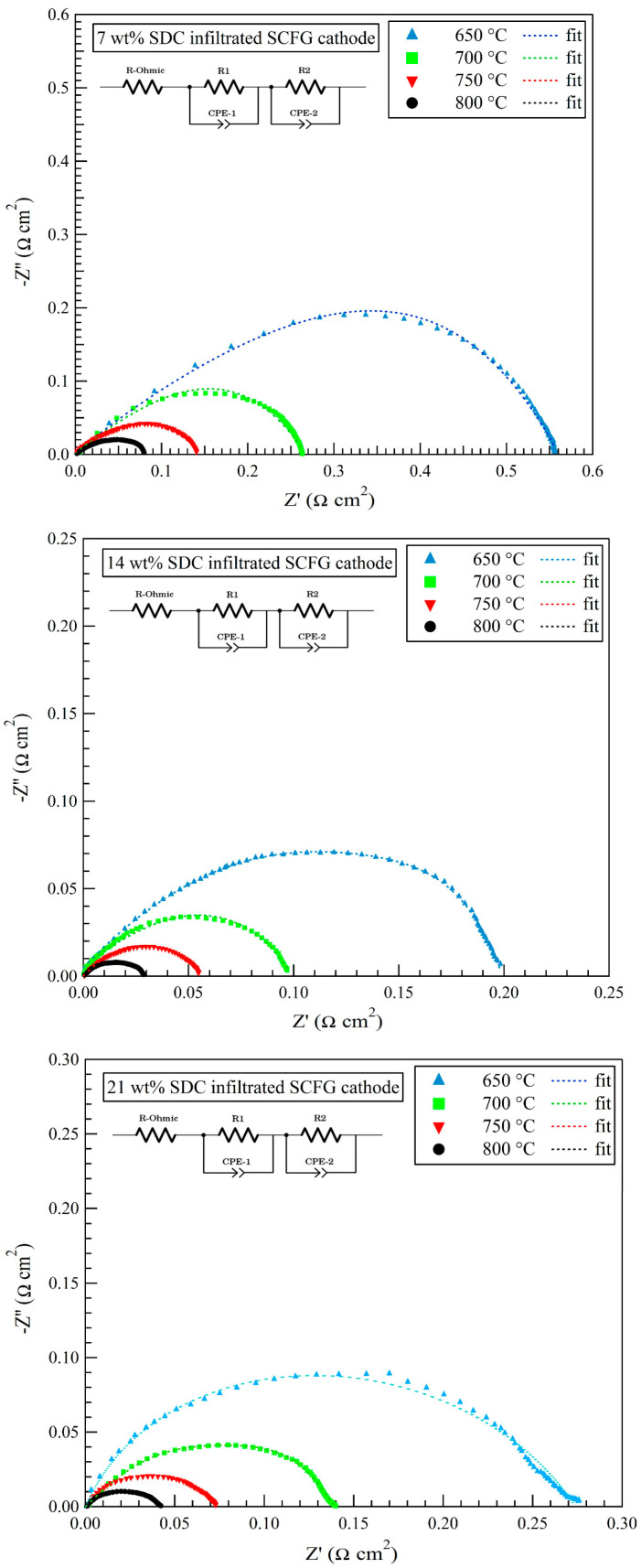
Nyquist impedance spectrum of the pure fibrous SCFG cathode and SDC-infiltrated SCFG electrodes with various amounts of SDC nanoparticles on an SDC electrolyte.

**Figure 6 materials-16-00399-f006:**
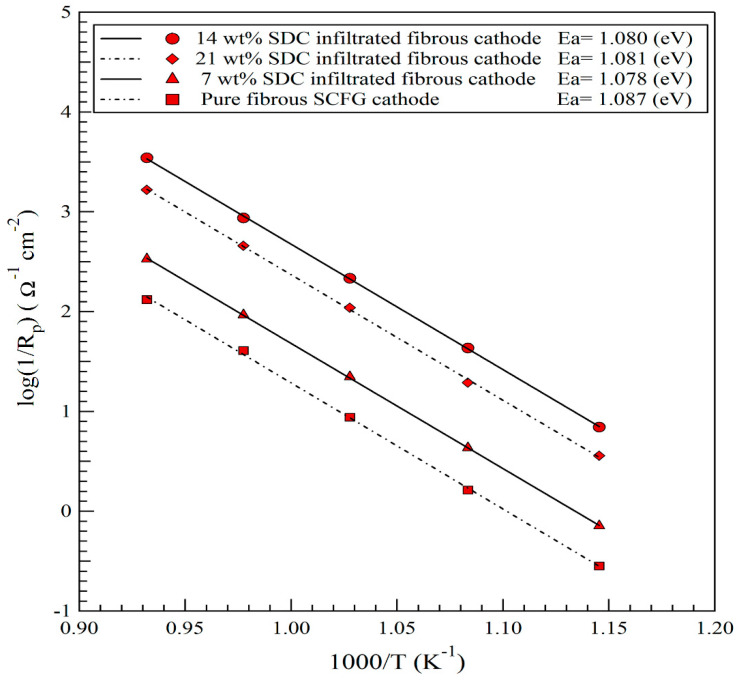
Arrhenius curve of polarization resistance of various fibrous SCFG cathodes, including pure and infiltrated with about 7, 14, and 21 wt% SDC nanoparticles.

**Table 1 materials-16-00399-t001:** Values of total polarization resistance and fitting parameters of EIS spectra for the various fibrous cathodes in the 650–800 °C temperature range.

Cathode	T (°C)	R_p_	R1 ^1^	CPE-1		R2 ^4^	CPE-2	
		(Ω cm^2^)	(Ω cm^2^)	Y_0_ ^2^ × 10^−3^	n ^3^	(Ω cm^2^)	Y_0_ ^5^ × 10^−2^	n ^6^
				(F^−1^ cm^−2^ s^n^)			(F^−1^ cm^−2^ s^n^)	
Fibrous SCFG	800	0.12	0.03	3.69	0.99	0.09	25.91	0.68
	750	0.2	0.07	0.05	0.99	0.11	9.92	0.57
	700	0.39	0.14	0.74	0.99	0.24	3.48	0.66
	650	0.81	0.22	0.01	0.96	0.52	3.23	0.45
7 wt% SDC infiltrated fibrous SCFG	800	0.08	0.02	4.82	0.99	0.06	55.26	0.62
	750	0.14	0.05	33.33	0.73	0.09	18.42	0.73
	700	0.26	0.08	0.05	0.98	0.17	16.5	0.58
	650	0.53	0.12	4.71	0.37	0.42	0.002	0.99
14 wt% SDC infiltrated fibrous SCFG	800	0.03	0.01	0.15	0.99	0.02	30.47	0.99
	750	0.05	0.02	244.3	0.61	0.03	0.06	0.8
	700	0.1	0.03	13.29	0.99	0.07	8.04	0.98
	650	0.19	0.07	0.72	0.99	0.12	4.81	0.74
21 wt% SDC infiltrated fibrous SCFG	800	0.04	0.03	13.9	0.98	0.01	30.08	0.72
	750	0.07	0.05	4.12	0.99	0.02	40.98	0.76
	700	0.13	0.08	0.09	0.99	0.04	1.04	0.8
	650	0.28	0.16	2.37	0.93	0.09	18.06	0.54

^1^ The standard deviation varied between 2% and 6%. ^2^ The standard deviation varied between 3% and 8%. **^3^** The standard deviation varied between 4% and 9%. ^4^ The standard deviation varied between 5% and 10%. ^5^ The standard deviation varied between 6% and 10%. ^6^ The standard deviation varied between 3% and 5%.

## Data Availability

The data presented in this study are available on request from the corresponding author.

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
