# Peer review of "Improved Electrochemical Performance of Sm0.2Ce0.8O1.9 (SDC) Nanoparticles Decorated SrCo0.8Fe0.1Ga0.1O3−δ (SCFG) Fiber, Fabricated by Electrospinning, for IT-SOFCs Cathode Application"

_materials, 2023, doi:10.3390/ma16010399_

Round 1

Reviewer 1 Report

This MS described a novel composite cathode decorated with SDC nanoparticles. The results of the phase and thermal analyses show interest and this work could be published after a minor revision.

1. Please describe each step of the TG analysis.

2. Please provide the phases and planes in Figure 2.

3. The papers https://doi.org/10.1016/j.chemosphere.2020.126966 and https://doi.org/10.1016/j.ijhydene.2019.11.241.

4. Please modify the conclusion to be more representative and clearly demonstrate the work.

Best regards,

Author Response

Reviewer #1:

This MS described a novel composite cathode decorated with SDC nanoparticles. The results of the phase and thermal analyses show interest and this work could be published after a minor revision.

  1. Please describe each step of the TG analysis.

Our response:

Thank you for your comments, as it has been highlighted in the article, each step of the TG analysis has been described in the text.

  1. Please provide the phases and planes in Figure 2.

Our response:

The phases have been carefully indicated in Fig. 2.  As the presented phases are well known, it maybe not need to provide planes for them on the figure, as it may cause the figure to be so crowded. Moreover, this kind of information can easy be found in references.

  1. Thepapers https://doi.org/10.1016/j.chemosphere.2020.126966 and https://doi.org/10.1016/j.ijhydene.2019.11.241.

Our response:

Thank you for your suggestion, but it should be noted that in the present work we tried to cite research works that are exactly related to the materials, experiments method and application in the current study and as the introduced studies are not related to SOFCs application, they may not be exactly suitable to be cited in the present work.

  1. Please modify the conclusion to be more representative and clearly demonstrate the work.

Our response:

Thank you for your comment. As you kindly suggested the conclusion is modified to be more representative.

Reviewer 2 Report

Comments on “ Improved electrochemical performance of Sm0.2Ce0.8O1.9 (SDC)  nanoparticles decorated SrCo0.8 Fe0.1 Ga0.1 O3- (SCFG) fiber, fabricated by electrospinning, for IT-SOFCs cathode application”

This paper is well-written and it is suitable for publication after minor improvement. 

1) This paper harnesses the electrospinning method to fabricate the needed fibers, however, the method has shortcomings in low output . Some advanced methods, e.g., the bubble electrospinning, should be briefly introduced. 

2) The nanoparticles can greatly enhance thermal conduction, for example, the carbon nanotube-embedded boundary layer theory for energy harvesting is widely used in engineering. 

3) English errors are not allowed for such an esteemed journal. 

Author Response

Reviewer #2:

Comments on “Improved electrochemical performance of Sm0.2Ce0.8O1.9 (SDC)  nanoparticles decorated SrCo0.8 Fe0.1 Ga0.1 O3-d (SCFG) fiber, fabricated by electrospinning, for IT-SOFCs cathode application” . This paper is well-written and it is suitable for publication after minor improvement.

  1. This paper harnesses the electrospinning method to fabricate the needed fibers, however, the method has shortcomings in low output. Some advanced methods, e.g., the bubble electrospinning, should be briefly introduced.

Our response:

Thank you for your suggestion, but as in this work conventional electrospinning method has been used for fabrication of SCFG fibers, we just introduced the works in which this method have been used for fabrication of SOFC cathode. In the other words, the aim of this work was not to introduce and compare different types of electrospinning methods and just the conventional method that has been introduced for preparing fibers for fabrication of SOFCs cathode, was intended.

  1. The nanoparticles can greatly enhance thermal conduction, for example, the carbon nanotube-embedded boundary layer theory for energy harvesting is widely used in engineering.

Our response:

That is right, the nanoparticles may greatly enhance thermal conduction, although the chemical composition of nanoparticles used has a very decisive role in improving the desired properties. By the way in the present study, we used SDC nanoparticles to enhance electrocatalytic activity and the results proved that it was successful.

  1. English errors are not allowed for such an esteemed journal.

Our response:

The manuscript has been reviewed carefully again and the English errors were corrected.

Reviewer 3 Report

The submitted manuscript reports on a electrospun material (SrCo0.8Fe0.1Ga0.1O3- (SCFG)) and its subsequent modification with Sm0.2Ce0.8O1.9 (SDC) for SOFC cells.  In general, the writing and language is ok. The quality of the described methodology and results is good. However, I have some comments to be addressed:

1.      Can the authors elaborate in the parameters used for the impedance measurements? They used 40 mV as perturbation amplitude, which is a little higher than the usual (4-times).

2.      The authors present EIS data which is the experimental data but not the fitting, although they present the Rp (polarization resistance) values. Can they show the fitting values for each of the elements fitted, as well as the goodness of fit? Please include the fit plots in figure5.

3.      In page 12, line 246, the sentence “Indeed, as the temperature rises, the concentration of oxygen vacancy is increased” requires a proper reference cited.

4.      In page 12, the sentence that starts in line 261 an refers to oxygen conduction properties of the cathode. This is aspect must be supported with the information obtained from the constant-phase element 2 (CPE2) as this is associated with oxygen transport phenomena. Also, please elaborate this part and match it with the EIS values. Also, please indicate why is CPE preferred instead of a capacitor.

Author Response

Reviewer #3:

The submitted manuscript reports on a electrospun material (SrCo0.8Fe0.1Ga0.1O3- (SCFG)) and its subsequent modification with Sm0.2Ce0.8O1.9 (SDC) for SOFC cells.  In general, the writing and language is ok. The quality of the described methodology and results is good. However, I have some comments to be addressed:

  1. Can the authors elaborate in the parameters used for the impedance measurements? They used 40 mV as perturbation amplitude, which is a little higher than the usual (4-times).

Our response:

Thank you for your comments. Regarding different references, it can be recognized that both 10 and 40 mV has been introduced for perturbation amplitude. In the other words, it does not make a significant change in the results of the impedance analysis.

  1. The authors present EIS data which is the experimental data but not the fitting, although they present the Rp (polarization resistance) values. Can they show the fitting values for each of the elements fitted, as well as the goodness of fit? Please include the fit plots in figure 5.

Our response:

Thank you for your comments. The fit plots are included in Fig. 5.

  1. In page 12, line 246, the sentence “Indeed, as the temperature rises, the concentration of oxygen vacancy is increased” requires a proper reference cited.

Our response:

Thank you for your comments. The following reference is added;

  1. Zhu, Y. Shi, C. Aruta, and N. Yang, Improving Electronic Conductivity and Oxygen Reduction Activity in Sr-Doped Lanthanum Cobaltite Thin Films: Cobalt Valence State and Electronic Band Structure Effects, ACS Appl. Energy Mater, 1 (2018) 5308–5317.

  1. In page 12, the sentence that starts in line 261 an refers to oxygen conduction properties of the cathode. This is aspect must be supported with the information obtained from the constant-phase element 2 (CPE2) as this is associated with oxygen transport phenomena. Also, please elaborate this part and match it with the EIS values. Also, please indicate why is CPE preferred instead of a capacitor.

Our response:

Thank you for your comments. In the mentioned paragraph we discussed about the effect of adding SDC nanoparticles on the cathode catalytic activity. We simply relate the increase in catalytic activity to the improvement in oxygen conductivity due to presence of SDC nano particles. Although we believe that the constant-phase element 2 is related to oxygen transport phenomena, but regarding the obtained Nyquist impedance spectrum, the relationship cannot be easily explained.

Regarding your question about why using CPE is preferred, please not that the curve obtained for oxygen transport phenomena is a depressed semicircle and cannot be fitted if a capacitor is chosen. 

Round 2

Reviewer 3 Report

The authors have worked and polished several aspects in the manuscript reflecting an increase in the overall quality. I am still having questions regarding the EIS analyses that would require to be addressed.

1. Previously, the authors were requested to present the values resulting from the fitting of the EIS spectra. To this point, the authors included the fitting itself in the plots, but the values from the fitting (Rp error %, Rs, CPE1, CPE2, etc) and the goodness of the fit (Chi-sqr) haven't been included. This is particularly important, as the Rp values presented are not yet related to a particular electrical circuit element such as R1 or R2. The previous argument also finds a discrepancy when the Rp value seems to be the intersection of the lowest frequency with the Z'-axis, which implies that Rp, apparently, results from the sum of R1 and R2 and invalidating the use of a 5-element electrical circuit for the fitting. Thus, Rp has to be one of the two R's while both resistances, R1 and R2, require a value to be reported. In this regard, authors are requested to include this data.

2. As a follow-up of the previous question, when requested to explain about the use of CPE elements instead of simple capacitor, the authors replied that the semicircle is depressed and cannot be fitted by a capacitor or when frequency dispersion is present (which requires it own justification). This would be valid if a single capacitor was used to fit the spectrum. However, there are two couples of elements in parallel, meaning there should be two semicircles, very possibly convoluted, as observed from the features in the spectra. Similarly to question 1, these both CPE elements require a value to be reported. Otherwise, CPE would have been arbitrarily used simply to fit the spectra (for mathematical purposes) and not to give meaning to the phenomena involved. Authors are requested to describe further this particular situation.
